# Women's preferences for caesarean or vaginal birth with a perspective of future fertility: A discrete choice experiment

**James D. Crispin** [1]*, **Ben W. Mol** [1,2], **Madelon van Wely** [3], **Daniel L. Rolnik** [1,4]

1 Department of Obstetrics and Gynaecology, Monash University, Clayton, Victoria, Australia, 2 Aberdeen Centre for Women's Health Research, University of Aberdeen, Aberdeen, United Kingdom, 3 Centre for Reproductive Medicine, Amsterdam UMC, University of Amsterdam, Amsterdam, The Netherlands, 4 Monash Women's, Monash Health, Clayton, Victoria, Australia

* JCri0002@student.monash.edu

**Data Availability Statement:** Deidentified raw data is included in Supporting Information.

**Funding:** The author(s) received no specific funding for this work.

## Abstract

### Objective

To investigate pregnant women's preferences for risks of vaginal and caesarean birth, including possible impacts on future fertility.

### Methods

In this discrete choice experiment, low-risk nulliparous pregnant women recruited after 28 weeks of gestation evaluated eight choice sets, each between two different hypothetical births scenarios which intermixed the risks of planned caesarean or vaginal birth. Scenarios consisted of six attributes: pain, maternal health, neonatal health, risk of unplanned intervention, impact on fertility and risk of complications in the next pregnancy. All scenarios contained risks to neonatal health as neither vaginal nor caesarean birth guarantee an ideal outcome. Choice data were analysed using a conditional logistic regression model.

### Results

Between June and September 2023, 211 participants, including 34 from pilot interviews, completed the questionnaire. Influential attributes were maternal health (conditional odds ratio [COR] 1.29, 95% CI 1.17 to 1.42, p<0.001) and risk of unplanned intervention (COR 1.37, 95% CI 1.24 to 1.51, p<0.001), favouring caesarean birth. Conversely, impact on fertility (COR 0.75, 95% CI 0.68 to 0.83, p<0.001) and complications in the next pregnancy favoured vaginal birth (COR 0.90, 95% CI 0.82 to 1.00, p = 0.045).

### Conclusions

Participants weighed the included morbidity risks of planned caesarean and vaginal birth in a low-risk pregnancy approximately equally. To facilitate an informed birth decision, clinicians should, apart from neonatal outcomes, particularly consider discussing impacts on fertility, maternal health and the risks of unplanned intervention or future pregnancy complications.

**Competing interests:** I have read the journal's policy and the authors of this manuscript have the following competing interests: BWM is supported by a NHMRC Investigator grant (GNT1176437). BWM reports consultancy, travel support and research funding from Merck and consultancy for Organon and Norgine. BWM holds stock from ObsEva. The remaining authors, JDC, DLR and MvW declare no competing interests. This does not alter our adherence to PLOS ONE policies on sharing data and materials.

## Introduction

Low-risk pregnant women are not routinely offered a caesarean because of the medical consensus that the risks outweigh the benefits [1–3]. However, many women may prefer the risk profile of a caesarean birth to a vaginal delivery [4–6].

In a low-risk pregnancy, it is unclear if planned caesarean or vaginal birth result in differences in maternal or perinatal mortality, and both are considered safe [7–10]. Caesarean birth carries higher risks of maternal puerperal infection or hysterectomy, the neonate developing childhood asthma or obesity, and complications in future pregnancies, particularly placenta accreta or uterine rupture [8, 11–13]. Recovery is longer, and emerging evidence suggests that caesarean delivery reduces subsequent live birth rate, although this is not routinely discussed with patients because it is unclear if it would affect their decisions [14–17]. Contrastingly, vaginal delivery is more painful acutely, carries higher risks of neonatal brachial plexus injury or hypoxic-ischaemic encephalopathy, and maternal pelvic organ prolapse or urinary incontinence [16]. Most women develop a perineal tear or require episiotomy [17, 18].

Existing birth preference research is mostly from qualitative studies which ask women what general factors matter to them or how they viewed previous birth experiences, so it is unclear how patients weigh the quantifiable morbidity risks of childbirth, which risks should be routinely disclosed when obtaining consent for caesarean delivery and if caesarean birth should be routinely offered in low-risk pregnancies [19–23].

This study aimed to investigate how nulliparous women in low-risk pregnancies preference the risks of vaginal and caesarean birth, including possible risks to their future fertility.

## Methods

This discrete choice experiment (DCE) involved participants making a series of choices between hypothetical birth scenarios such that the choice data generated could be analysed to determine which attributes or risks of the scenarios were most influential in their decisions. It was designed in accordance with discrete choice experiment good research practice guidelines published by The Professional Society for Health Economics and Outcome Research (IPSOR) [24].

### Ethics approval

Ethical approval was obtained from the Monash Health Human Research Ethics Committee (RES-23-0000072A) on the 18th of May 2023, and the study was registered with the Monash University Human Research Ethics Committee (project ID: 38609).

### Setting and participants

This study was performed in four teaching hospitals all working under the umbrella of Monash Health in Melbourne, Australia, between June 1st and September 7th, 2023. Antenatal clinic records were screened to determine if patients met the inclusion criteria.

Women were invited to participate via text, email, leaflets, and in-person in antenatal clinic waiting rooms if they were nulliparous, in a low-risk pregnancy and had completed their 28-week antenatal visit. The exclusion criteria were age under 18 years, English illiteracy, or any non-low risk pregnancy, which was defined by the presence of gestational diabetes, BMI $\geq$43 kg/m$^2$ or any medical conditions requiring specialist obstetric or medical management at the time of survey completion. The BMI cut-off of 43 kg/m$^2$ was chosen as women above this require specialist obstetric care at our institution. Electronic written informed consent was obtained from all participants, and women were not provided with financial or other incentives to participate.

To estimate the sample size required, we used a rule-of-thumb method proposed by Lancsar and Louviere (2008) which suggests 20 responses per attribute, yielding a sample size estimate for this study of 120 participants [25].

## Attributes and levels

Each of the hypothetical birth scenarios was described based on six different attributes (Table 1), as existing literature indicates that participants may become overwhelmed if more are included [26]. Each attribute was described in lay terms, and lay definitions were provided for any necessary medical jargon (e.g., placenta accreta) at the start of the questionnaire. Future fertility, or the 'chance of having another baby in the future', was included as an attribute to determine if reduced live birth rate after caesarean section is likely to affect participants' birth decisions, and hence if it should be included in consent discussions. A literature search was conducted to determine the remaining five attributes.

Each attribute was designed with two possible levels, representing the risks of an elective caesarean, or planned vaginal birth in a low-risk pregnancy. Included risks were grouped into three categories of magnitude (small, medium, and large; Fig 1) based on the number of additional women or neonates affected, which was determined from existing literature (Table 2) [27]. When an absolute risk difference was not available, the risk descriptor was chosen using the general population incidence and relative risk difference.

Among the existing, largely qualitative research on pregnant women's birth preferences, 'health of the baby' has consistently ranked as the most important factor [20]. The risks of developing brachial plexus injury and hypoxic-ischemic encephalopathy were included in the level of this attribute representing vaginal birth, and the risks of neonatal respiratory morbidity and developing asthma or obesity in childhood were included to represent caesarean birth [11, 12, 28–30].

Health of the mother is also highly prioritised by pregnant women in prior research and was therefore included as an attribute [20]. The risks of needing an episiotomy or developing urinary incontinence or perineal tears were included in the level representing vaginal birth, while an increased risk of puerperal infection and longer recovery time, including an inability to lift heavy items or drive for 2–6 weeks after delivery, were included in the level representing caesarean birth [8, 13, 18, 33].

While most existing preference research has not asked pregnant women to consider the risk of complications in future pregnancies, these risks are included in consent guidelines and may represent an extension of safety of the mother and neonate [1, 2]. Therefore, the two major future pregnancy risks of placenta accreta and uterine rupture were included collectively as an attribute.

**Table 1. Attributes and levels used in the discrete choice experiment.**

| Attribute | Levels | |
|---|---|---|
| | Representing caesarean birth | Representing vaginal Birth |
| **Pain** | None during delivery, more while recovering. | Significant during delivery, less while recovering. |
| **Health of the mother** | Small risk of infection. Longer hospital stay, cannot lift heavy items or drive for 2–6 weeks. | Large risk of urinary incontinence. Most mothers will get a perineal injury. |
| **Health of the baby** | Medium risk of breathing difficulties and/or developing asthma or obesity in childhood. | Small risk of arm paralysis and/or brain injury. |
| **Risk of extra (unplanned) intervention** | Low | High |
| **Chance of having another baby in the future** | Medium reduction | No change |
| **Risk of placenta accreta or uterine rupture in next pregnancy** | Small increase | No change |

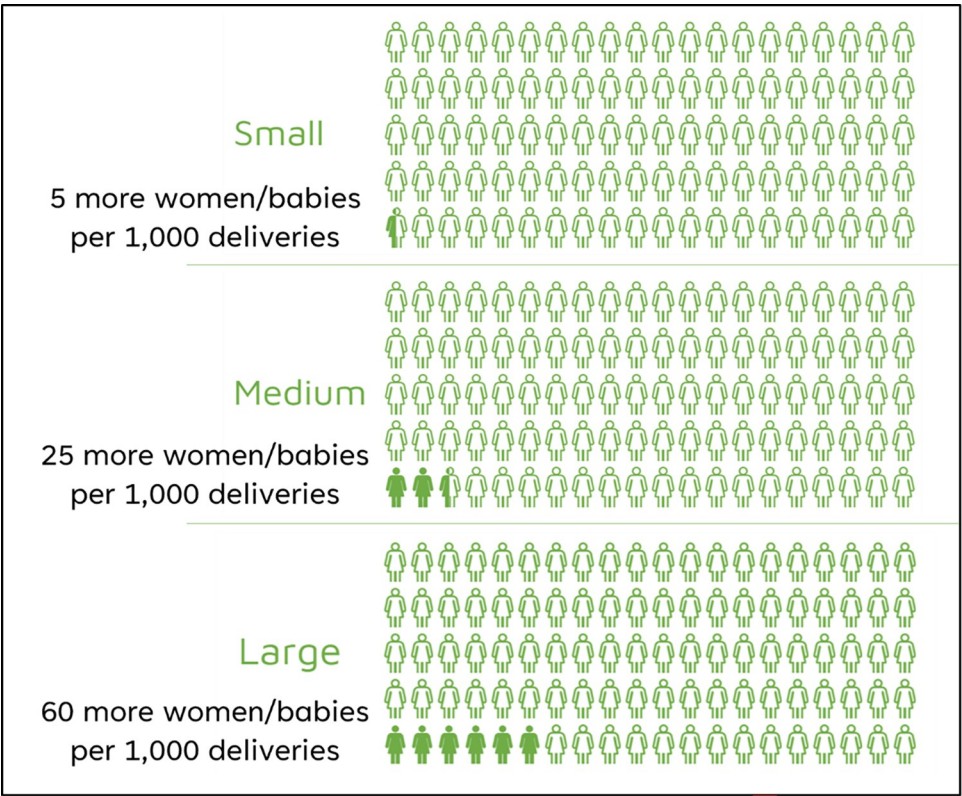

**Fig 1. Absolute risk differences infographic used in the questionnaire.** Created using LiveGap Charts [27].

Pain was also an attribute of the DCE as it has been identified as a major deciding factor between caesarean and vaginal birth [19, 20].

A greater sense of control may be an important factor driving women to request a caesarean and was included as the final attribute [21–23]. In the scenarios, this was expressed as the 'Risk of extra (unplanned) intervention', with 'high' representing higher risk of unexpected medical intervention with vaginal birth and 'low' representing the more predictable nature of planned caesarean delivery. These levels do not have a statistical basis as there is no established method to quantify differences in uncertainty between caesarean and vaginal birth.

### Choice set design

Eight choice sets each containing two mirrored (opposite) alternatives (Fig 2) were included in the DCE. The single-block design was produced using Version 1.2.1 of the 'AlgDesign' package for R software (Version 4.2.1) which randomly mixed risks of caesarean and vaginal birth (levels) between hypothetical birth scenarios to produce an orthogonal and balanced fractional factorial design consisting of eight choice sets [34–36].

### Questionnaire design

The questionnaire was administered electronically via the Qualtrics® survey platform (Qualtrics, Provo, UT), and a complete example is found in S1 File [37]. The first part collected demographic covariates, including age, gestational age, occupation, country of birth, ethnicity, highest level of education attainment, history of infertility treatment complications in the current pregnancy, expected mode of delivery, if their doctor or midwife had discussed the option

**Table 2. Risk descriptors denoting the absolute difference in the probability of each medical risk between caesarean and vaginal birth in the questionnaire.**

| Outcome | Absolute risk difference per 1,000 deliveries (95% confidence interval) | Relative risk or odds ratio (95% confidence interval) | General population incidence | Risk descriptor |
|---|---|---|---|---|
| Urinary Incontinence [13] | -60 (-45 to -71) | N/A | N/A | Large |
| Major puerperal infection [8] | 4.3 (3.6 to 5.1) | N/A | N/A | Small |
| Brachial plexus injury [28] | Unavailable | OR 0.15 (0.13 to 0.18) | 1.24 cases per 1,000 live births | Small |
| Hypoxic ischaemic encephalopathy [29] | Unavailable | OR 0.17 (0.05 to 0.56) | 3.8 cases per 1,000 live births | Small |
| Respiratory morbidity in term babies [30] | Unavailable | RR 1.9 (1.2 to 3.0) | 18 cases per 1,000 live births | Medium |
| Development of childhood asthma [11, 31] | Unavailable | RR 1.20 (1.15 to 1.25) | 100 cases per 1,000 children | Medium |
| Development of childhood overweightness or obesity [12, 32] | Unavailable | RR 1.10 (1.01 to 1.18) | 250 cases per 1,000 children | Medium |
| Live birth rate [17] | 30 | N/A | N/A | Medium |
| Placenta accreta in future pregnancy [13] | 0.57 (0.30 to 0.96) | N/A | N/A | Medium |
| Uterine rupture in future pregnancy [13] | 9.82 (3.97 to 23.32) | N/A | N/A | |

RR: Relative risk; OR: Odds ratio.

Where the absolute risk difference has not been reported in existing literature, an estimate was made based in on the relative risk and general population incidence. In cases of uncertainty, the risk descriptor was selected to favour the null hypothesis, that women prefer the risks associated with vaginal birth. An estimate for the combined risk of placenta accreta and uterine rupture was obtained by summing the point estimates of each risk individually.

of a caesarean section with them and if they had plans for more children after their current pregnancy.

Participants were shown lay definitions of the risks discussed in the hypothetical scenarios and guided through an example choice set. Then, participants were asked to answer nine choice sets, the eight choice sets which comprised the fixed DCE design, and a repeat choice set (consistency test), included to test whether participants gave consistent responses. The

| Attribute | Birth Scenario A | Birth Scenario B |
|---|---|---|
| **Pain** | No pain during delivery, more while recovering. | Significant pain during delivery, less while recovering. |
| **Health of the mother after delivery** | Large risk of urinary incontinence. Most mothers will get a perineal injury. | Small risk of infection. Longer hospital stay, cannot lift heavy items or drive for 2-6 weeks. |
| **Health of the baby after delivery** | Small risk of arm paralysis and/or brain injury. | Medium risk of breathing difficulties and/or developing childhood asthma or obesity. |
| **Risk of extra (unplanned) intervention** | Low | High |
| **Chance of having another baby in the future** | Medium reduction | No change |
| **Risk of placenta accreta or uterine rupture in next pregnancy** | Small increase | No change |

**Fig 2. Example of a choice set used in the DCE questionnaire.**

order of the choice sets was randomised for each participant using the Qualtrics® question randomisation function.

Finally, participants were asked to describe any difficulties they encountered during the questionnaire and if they had any suggestions for improvement in two free-text answer questions.

### Data collection

Data were collected in two phases. Pilot interviews involving the author (JDC) guiding participants through the questionnaire on the Qualtrics® platform were first performed to ensure it was manageable and participants understood the risks as described. Encountered difficulties and suggestions for improvement reported in the two free-text questions at the end of the questionnaire were analysed by JDC using the content analysis method described by Elo and Kyngäs (2008) [38].

Subsequently, the questionnaire was distributed online for participants to complete independently, using the recruitment methods described above.

### Statistical analysis and outcome

Differences in demographics between the pilot interview and online survey participants were analysed for statistical differences. Continuous variables were analysed using a two-sided independent t-test and normality of the underlying distribution was confirmed with the Shapiro-Wilk test, while homogeneity of variances was checked with Levene's test. Categorical variables were analysed using Pearson's Chi-Square test, or Fisher's exact test, as appropriate.

The choice data was dummy coded using the level for each attribute representing vaginal birth as a reference (0) compared to the level representing caesarean birth (1). Due to the multiple choice sets answered by each participant and the expected within-subject correlation, the formatted data was then analysed using a conditional logistic regression model, which produced the conditional log odds coefficients for each of the attributes in the hypothetical scenarios [39]. The model was fitted using R software (Version 4.2.1) and only accounted for main effects [36, 40].

A priori, we expected all attributes to be significant. The sign of the beta coefficient reflects whether the attribute has a positive or negative effect on preference score. To weigh the relative importance of the attributes, the beta coefficients of attributes with significant impact were compared.

Two sensitivity analyses were performed, the first excluded participants who failed the consistency test and the second was limited to women aged 30 and older, as evidence for reduced fertility after caesarean birth is strongest in this population [17].

### Results

Between June and September 2023, we invited 702 pregnant women to complete the DCE, yielding 211 (30.1%) eligible responses, including thirty-four pilot interviews (S1 Fig). Participants were 30.5 years of age on average, only half (45.5%) identified as Australian, most (61.6%) had a university education, one quarter (24.2%) worked in the healthcare industry and most (64.9%) probably or definitely planned for more children in the future. Almost all expected a vaginal birth (96.2%) (Table 3).

In the descriptive analysis of difficulties reported by pilot interview participants, one third (29.4%) reported no difficulties, and almost half (47.1%) described difficulty choosing between the risks. Only one quarter of participants provided any suggestions for improvement (S2 Table), and the most common suggestion made by 9% of participants was to reduce the number of risks in each scenario. No changes were made to the choice sets following this feedback

**Table 3. Demographic characteristics of respondents meeting the inclusion criteria.**

| Characteristic | All Participants (N = 211) | Pilot interview participants (n = 34) | Online survey participants (n = 177) | p-value |
|---|---|---|---|---|
| Mean age (SD)–years (SD) | 30.5 (4.7) | 30.8 (3.9) | 30.5 (4.9) | 0.735 |
| Mean gestational age at completion (SD)–weeks | 32.2 (3.8) | 32.6 (3.8) | 32.2 (3.8) | 0.567 |
| Ethnicity–n. (%) | | | | 0.461 |
| Australian (non-Indigenous) | 96 (45.5%) | 17 (50.0%) | 79 (44.6%) | |
| South-east Asian | 50 (23.7%) | 8 (23.5%) | 42 (23.7%) | |
| Other | 65 (30.8%) | 9 (26.5%) | 56 (31.6%) | |
| Country of birth | | | | 0.853 |
| Australia | 99 (46.9%) | 16 (47.1%) | 83 (46.9%) | |
| India | 26 (12.3%) | 3 (8.8%) | 23 (13.0%) | |
| Other | 86 (40.8%) | 15 (44.1%) | 71 (40.1%) | |
| Highest level of education completed–n (%) | | | | 0.026* |
| Secondary school or vocational certificate | 81 (38.4%) | 7 (20.6%) | 76 (41.8%) | |
| University degree | 130 (61.6%) | 27 (79.4%) | 103 (58.2%) | |
| Industry of study or employment–n (%) | | | | 0.812 |
| Not employed or studying | 25 (11.8%) | 2 (5.9%) | 20 (11.3%) | |
| Healthcare and social assistance | 51 (27.4%) | 11 (34.4%) | 40 (26.0%) | |
| Other | 135 (64.0%) | 13 (38.2%) | 117 (66.1%) | |
| Received prior treatment for infertility–n (%) | 17 (8.1%) | 1 (2.9%) | 16 (9.0%) | 0.726 |
| Plans for more children after current pregnancy–n (%) | | | | 0.037* |
| Probably or definitely yes | 137 (64.9%) | 27 (79.4%) | 110 (62.1%) | |
| Might or might not | 52 (24.6%) | 3 (8.8%) | 49 (27.7%) | |
| Probably or definitely no | 22 (10.4%) | 4 (11.8%) | 18 (10.2%) | |
| Discussed caesarean birth with doctor or midwife before completing questionnaire–n (%) | 38 (18%) | 10 (24.9%) | 28 (15.8%) | 0.059 |
| Expected caesarean birth–n (%) | 8 (3.8%) | 0 (0.0%) | 8 (4.5%) | 0.360 |

Note that this table only shows the most common ethnicities, countries of birth and industries of study or employment.

as difficulty choosing between the numerous included risks may reflect the inherent complexity of the birth decision. However, the questionnaire was amended to include the risk difference infographic (Fig 1) on every page to help participants weigh the risks.

In the descriptive analysis of all online survey participants' feedback, most (72.9%) participants reported no difficulties. The most common feedback was difficulty concentrating on scenarios due to their similarity (11.3%) and difficulty choosing between the risks (7.9%). S1 and S2 Tables contain the full descriptive analysis of reported difficulties from both pilot interview and online survey participants.

In the primary analysis of all participants (Fig 3), beta coefficients and conditional odds ratios for the attributes were approximately symmetrically distributed between favouring a caesarean or vaginal birth. Patients weighted risk of an unplanned intervention as most important followed by risk of infertility, health of the mother and risk of placenta accreta. Participants preferred the risks to the health of the mother after delivery (conditional OR 1.29, 95% CI 1.17 to 1.42, p<0.001) and lower risk of extra (unplanned) intervention (conditional OR 1.37, 95% CI 1.24 to 1.51, p<0.001) associated with a caesarean section. Two attributes were weighed significantly in favour of a vaginal birth, 'chance of having another baby in the future' (conditional OR 0.75, 95% CI 0.68 to 0.83, p<0.001) and 'risk of placenta accreta or uterine rupture in next pregnancy' (conditional OR 0.90, 95% CI 0.82 to 1.00, p = 0.045). There was no statistically significant preference for the remaining two attributes in favour of caesarean or

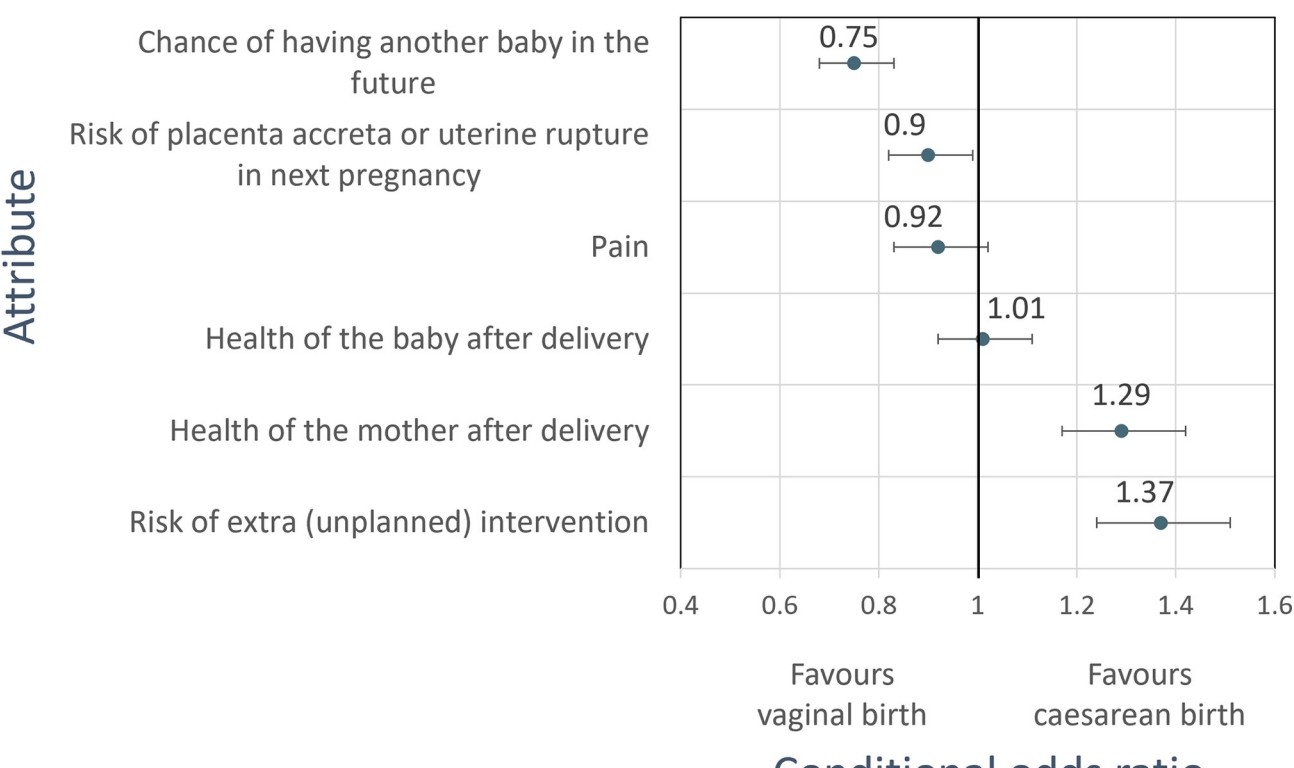

**Fig 3. Conditional odds ratios for each attribute from the primary analysis of all participants.** These represent the conditional odds of participants choosing the level of each attribute representing caesarean birth over the level representing vaginal birth.

vaginal birth. 'Pain' showed a non-statistically significant trend in favour of a vaginal delivery (conditional OR 0.92, 95% CI 0.83 to 1.01, p = 0.087), and 'health of the baby after delivery' was not predictive of participants' choices (conditional OR 1.01, 95% CI 0.92 to 1.12, p = 0.838).

When comparing the beta coefficients of significant attributes, perceived benefits to the health of the mother with a caesarean birth were 2.5 times more important to participants than the increased risk of placenta accreta or uterine rupture in the next pregnancy. Although the risk to fertility with a caesarean was 1.2 times more important than perceived maternal health benefits from the procedure, the lower risk of unplanned intervention with caesarean birth was 1.1 times more important than its impact on fertility.

Sensitivity analysis of pilot interview participants only (S3 Table) revealed a similar trend in preferences to the primary analysis, except for the 'health of the baby after delivery' attribute, which was weighed significantly in favour a caesarean birth (conditional OR, 1.38, 95% CI 1.05 to 1.77, p = 0.019). However, the interview and online survey samples were significantly different in terms of educational attainment (p = 0.026) and plans for future pregnancies (p = 0.037).

Sensitivity analysis of participants who passed the consistency test and participants aged 30 years and older revealed similar preferences to the primary analysis (S3 Table).

## Discussion

### Main findings

In this discrete choice experiment involving low-risk nulliparous pregnant women, there was no clear preference for a caesarean or vaginal birth when the major risks of each were weighed

collectively. Women showed a statistically significant preference in favour of the lower risk of unplanned intervention and the risks to maternal health with a caesarean birth. There was no established preference for risks to the health of the neonate in favour of a caesarean or vaginal birth. Future fertility was the second most influential factor in participants' decisions, favouring a vaginal delivery.

## Limitations

To our knowledge, this is the first study to quantitatively investigate how low-risk pregnant women preference their medical risks associated with a caesarean or vaginal birth, but it has some limitations.

Not all risks of caesarean and vaginal birth could be included, and some attributes may be asymmetrically balanced. Notably, the survey did not include potential differences in maternal or neonatal mortality, although the Royal College of obstetricians and Gynaecologists and its Australian counterpart quote an increase of two maternal and neonatal deaths per 10,000 planned caesarean deliveries, compared to a planned vaginal delivery [1, 2]. These figures were excluded because each are only informed by a single low-quality retrospective cohort study which may have biased results in favour of vaginal delivery by excluding women in the planned vaginal group who developed complications after 39 weeks that made their pregnancy high-risk [10, 12, 41]. However, even a small mortality difference may have an important influence on pregnant women's decisions.

Although the magnitudes of most risks were quantified based on prior literature, this could not be achieved for the 'pain' and 'risk of extra (unplanned) intervention' attributes. This is an important consideration given that the 'risk of extra (unplanned) intervention' was the most influential attribute in participants' decisions but may have been variably interpreted.

Given the 32% response rate, the survey likely suffered a self-selection bias which may have favoured women motivated to share their opinions, although the direction of this bias is unclear. One quarter of respondents also worked in the healthcare sector. Although the DCE was designed to mitigate against intrinsic biases favouring a caesarean or vaginal birth by not informing participants of the mode of delivery associated with each risk, healthcare workers may know these associations from their training or practice. The educational backgrounds of these participants may also have influenced their views, and further research should explore how risk preferences may differ in patients from different education backgrounds.

Furthermore, low-risk nulliparous women may weigh risks differently from high-risk pregnant women, or women with previous pregnancy experiences. This limits the generalisability of our findings and highlights an avenue for future research.

Pilot interview participants had a significantly higher level of education than online survey participants and significantly more expressed plans for future children. This may explain differences in how these groups weighed risks to the neonate. However, we cannot exclude that interview participants may have been better informed of the risks. As a more highly educated sample piloted the questionnaire, it is unclear how well the online survey participants understood the risks presented and this may limit validity of the online survey results. Finally, the sample size for this study was calculated using a rule-of-thumb, and therefore may not have been adequately powered to detect statistically significant preferences for some attributes.

## Interpretation

Our findings add to the existing body of research challenging the paradigm that, in a healthy pregnancy, a planned vaginal birth carries a better maternal risk profile than an elective caesarean, but instead suggests that this decision may be preference sensitive [42]. Further research

is needed to substantiate this finding. However, obstetricians should consider discussing the risks and benefits of an elective caesarean with their low-risk patients to ensure they are delivering patient-centred care.

It is important to note that the World Health Organization's optimal caesarean rates are based on the premise that in the absence of a clear mortality difference between a planned caesarean and vaginal birth, the morbidity difference is so great that a vaginal birth should be preferred in all cases [43, 44]. Our findings cast doubt on this premise, supporting calls to reduce reliance on generic population-wide caesarean rate targets in favour of individualised, preference-specific maternity care.

Our finding that the lower 'risk of extra (unplanned) intervention' with caesarean birth was the most influential factor in participant's decisions supports existing research which suggests that a sense of control is paramount to women's birthing decisions [21].

We also found that future fertility may be an important factor in low-risk pregnant women's birth decisions. There is ongoing debate about the magnitude of the risk to fertility after caesarean birth given that existing data are from retrospective cohort studies, many of which poorly account for confounders including maternal age and comorbidities [14, 16, 17, 45]. However, given our findings, obstetricians should consider informing their patients of the available evidence for this risk and its limitations.

Pain is frequently cited as a factor driving women to request a caesarean birth but did not have a significant effect on our participants' decisions [20]. This finding may reflect that most participants expected a vaginal delivery. Additionally, existing research focused on delivery pain, but we considered both delivery and recovery pain [20]. Therefore, our findings underline the importance of informing patients of the worse recovery pain after a caesarean delivery.

## Conclusion

In this DCE involving low-risk pregnant women choosing between hypothetical birth scenarios containing unique combinations of the risks of planned caesarean or vaginal birth, health of the mother, risk of unplanned intervention, impact on fertility and complications in future pregnancies had significant effects on participants' decisions. Apart from neonatal outcomes, clinicians should particularly consider including these factors in consent discussions to facilitate informed birth choices. As participants weighed the included morbidity risks evenly between favouring caesarean or vaginal delivery, obstetricians should consider discussing the risks and benefits of an elective caesarean with low-risk patients as mode of birth may be preference-sensitive for these women.

## Supporting information

**S1 File. DCE questionnaire.**
(DOCX)

**S2 File. Raw data.**
(XLSX)

**S1 Fig. Recruitment flow chart. LGA: large for gestational age.**
(TIF)

**S1 Table. Content analysis of difficulties reported as free text by respondents. Note that some participants reported multiple difficulties.**
(DOCX)

**S2 Table. Content analysis of suggestions for improvement reported as free text by respondents. Note that some participants provided multiple suggestions.**
(DOCX)

**S3 Table. Conditional odds ratios for each attribute for all participants, interview participants only, participants who passed the consistency test only and participants ≥30 years of age only.**
(DOCX)

## Acknowledgments

We would like to thank all the pregnant women who donated their time and attention to this study. We also acknowledge and thank all the staff across Monash Health who facilitated patient recruitment for this study.

## Author Contributions

**Conceptualization:** James D. Crispin, Ben W. Mol, Daniel L. Rolnik.

**Data curation:** James D. Crispin, Madelon van Wely, Daniel L. Rolnik.

**Formal analysis:** James D. Crispin, Madelon van Wely, Daniel L. Rolnik.

**Investigation:** James D. Crispin.

**Methodology:** James D. Crispin, Ben W. Mol, Madelon van Wely, Daniel L. Rolnik.

**Project administration:** James D. Crispin.

**Supervision:** Ben W. Mol, Madelon van Wely, Daniel L. Rolnik.

**Writing – original draft:** James D. Crispin.

**Writing – review & editing:** James D. Crispin, Ben W. Mol, Madelon van Wely, Daniel L. Rolnik.

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
