## [Decision Letter · Decision Letter 0]

11 Jul 2024

PONE-D-24-08282Women’s Preferences for Caesarean or Vaginal Birth with a Perspective of Future Fertility:

A Discrete Choice ExperimentPLOS ONE

Dear Dr. Crispin,

Thank you for submitting your manuscript to PLOS ONE. After careful consideration, we feel that it has merit but does not fully meet PLOS ONE’s publication criteria as it currently stands. Therefore, we invite you to submit a revised version of the manuscript that addresses the points raised during the review process.

Please include the following items when submitting your revised manuscript:A rebuttal letter that responds to each point raised by the academic editor and reviewer(s). You should upload this letter as a separate file labeled 'Response to Reviewers'.A marked-up copy of your manuscript that highlights changes made to the original version. You should upload this as a separate file labeled 'Revised Manuscript with Track Changes'.An unmarked version of your revised paper without tracked changes. You should upload this as a separate file labeled 'Manuscript'.If applicable, we recommend that you deposit your laboratory protocols in protocols.io to enhance the reproducibility of your results. Protocols.io assigns your protocol its own identifier (DOI) so that it can be cited independently in the future. For instructions see: https://journals.plos.org/plosone/s/submission-guidelines#loc-laboratory-protocols. Additionally, PLOS ONE offers an option for publishing peer-reviewed Lab Protocol articles, which describe protocols hosted on protocols.io. Read more information on sharing protocols at https://plos.org/protocols?utm_medium=editorial-email&utm_source=authorletters&utm_campaign=protocols.

We look forward to receiving your revised manuscript.

Kind regards,

Natasha L Pritchard

Academic Editor

PLOS ONE

Journal Requirements:

2. Thank you for stating the following in the Competing Interests section: "I have read the journal's policy and the authors of this manuscript have the following competing interests: BWM is supported by a NHMRC Investigator grant (GNT1176437). BWM reports consultancy, travel support and research funding from Merck and consultancy for Organon and Norgine. BWM holds stock from ObsEva. The remaining authors, JDC, DLR and MvW declare no competing interests."

3. In the online submission form, you indicated that "Deidentified raw data is available is a single spreadsheet upon request from the author. This was a requirement of our human research ethics committee given the sensitive nature of some of the included questions."

Additional Editor Comments:

This is a well-written, patient-centred study that provides insight into which risks matter to mothers. This is a very valuable question in the current clinical landscape in which patient autonomy and informed consent about all birth options is increasingly valued. The study design was well thought out, and provided a blinded approach to risk associated with each choice. I thoroughly enjoyed reading this novel research, thank you.

Comments:

-It is great that future fertility has been included as a factor because, as shown by this study, it can be a very concerning and influential factor for women when informing their birth preferences

-I was not able to work out exactly where the “reduction in fertility” becoming a “medium reduction” came from. Is that the 3.0% in table S1 (which is 30 per women/ babies and therefore ‘medium’ risk by the diagram?) If so, why was ‘major puerperal infection’ (at 4.3 per 1000) not listed as a medium risk?

-When reading the major study cited for reduction in fertility after CS birth, it appears that a lot of the risk could be due to indications for the caesarean in the first place. Regardless, this discrete choice experiment is important in showing the value to the women of potential future fertility.

-It would be ideal if S1 could be included in the main text

-Minor – line 66 – should be “because it is unclear if IT would affect their decisions.”

-The exclusion criteria are reasonable, but it may be that the excluded cohorts differentially value risks after their high-risk pregnancy experiences. I agree with the reviewer in that this could be an avenue for further research.

Reviewers' comments:

Reviewer's Responses to Questions

**Comments to the Author**

1. Is the manuscript technically sound, and do the data support the conclusions?

Reviewer #1: Yes

Reviewer #2: Yes

2. Has the statistical analysis been performed appropriately and rigorously? 

Reviewer #1: Yes

Reviewer #2: Yes

3. Have the authors made all data underlying the findings in their manuscript fully available?

Reviewer #1: No

Reviewer #2: Yes

4. Is the manuscript presented in an intelligible fashion and written in standard English?

Reviewer #1: Yes

Reviewer #2: Yes

5. Review Comments to the Author

Reviewer #1: This is a very interesting, clinically relevant discrete choice experiment exploring women’s risk preferences when considering a vaginal birth or caesarean section.

Comments as follows:

Were there financial or other incentives offered for women who participated?

Women preferred the risk profile of a caesarean section when considering unplanned intervention and maternal health; yet almost all women expected to have a vaginal delivery. Is long-term data available for these women? Is it known whether any of these women subsequently pursued a maternal request caesarean section or asked about this with their antenatal care providers?

The disproportionate number of healthcare workers is appropriately addressed as a limitation. The authors note that this bias is mitigated as patients were not informed of the mode of delivery associated with each risk in the experiment. However, many healthcare workers (I would think almost all participants with midwifery or obstetric backgrounds) would know the mode of delivery associated with each risk from clinical experience. It is therefore perhaps worth rephrasing lines 292-295.

Additionally, the health literacy of this cohort is likely to be much higher than the background population. It would be interesting to know whether the same risk preferences would be observed in a less educated background, and whether it would ultimately change birth preferences. This could be suggested as an avenue for future research.

Table 1: The text representing caesarean birth and vaginal birth under “Health of the baby” seem to be incorrect, ie. Medium risk of breathing difficulties and/or developing asthma or obesity in childhood should be listed as representing caesarean birth.

References are required for S1: “Major puerperal infection” and “Livebirth rate”.

Thank you for the opportunity to peer review this paper.

Reviewer #2: Summary

The paper investigates the preferences of low-risk nulliparous pregnant women regarding the risks associated with vaginal and caesarean births, using a discrete choice experiment (DCE) methodology. This study is significant because, despite the medical consensus that caesarean births should not be routinely offered due to higher associated risks, some women may prefer the risk profile of a caesarean birth. The research aims to quantify how these women weigh various risks, including pain, maternal and neonatal health, risk of unplanned intervention, impact on future fertility, and complications in future pregnancies.

Participants were recruited from four teaching hospitals under Monash Health. The DCE involved participants evaluating eight choice sets comparing different birth scenarios. The choice data were analysed using conditional logistic regression, revealing that maternal health and the risk of unplanned intervention significantly influenced a preference for caesarean birth, while the impact on fertility and risk of complications in future pregnancies favoured vaginal birth.

The findings highlight the importance of discussing not only neonatal outcomes but also fertility impacts, maternal health, and risks of unplanned interventions or future pregnancy complications in birth planning conversations. This approach can help ensure that women are fully informed and can make decisions aligned with their preferences, ultimately improving the quality of maternal care.

Methods

The methods section is detailed and clear, providing a thorough description of the study design, participant recruitment, attribute selection, choice set design, questionnaire administration, data collection, and statistical analysis. However, the following improvements can be considered:

- Clarify how the randomisation of the choice sets was handled to ensure balanced presentation across participants.

- Provide more detail on how the free-text responses from the pilot phase were used to refine the questionnaire.

- Discuss any potential biases or limitations in the sample selection and how they might impact the generalisability of the findings.

Results

It is reasonable to include p-values to highlight significant differences in baseline characteristics, especially when comparing pilot interview participants and online survey participants. This helps assess whether these differences might impact the study's results. However, the p-value of 0.05 is arbitrary and there is not much difference in terms of evidence strength between p=0.049 and p=0.051.

- Reporting p-values helps identify significant differences in baseline characteristics that might impact the study’s outcomes. However, consider discussing these differences in the context of their potential impact on the results.

- As an alternative to p-values, or to complement them, consider reporting standardised differences. This approach focuses on the magnitude of differences rather than just statistical significance.

- Report the magnitude of the measure of association, 95% CI, and p-value together throughout the paper (including Table S5).

- In Table S1, use the term “Relative Risk or Odds Ratio” for the third column.

6. PLOS authors have the option to publish the peer review history of their article (what does this mean?). If published, this will include your full peer review and any attached files.

Reviewer #1: No

Reviewer #2: No

---

## [Author Response · Author response to Decision Letter 0]

26 Aug 2024

Dear Dr Pritchard,

We would like to thank you and the reviewers for your valuable feedback. In our attached letter, we have described in detail the changes made in response to the suggestions provided. We hope our revised work meets your expectations. Thank you for your ongoing consideration of our manuscript.

Kind regards,

James Crispin

---

## [Editor Report · Decision Letter 1]

3 Sep 2024

Women’s Preferences for Caesarean or Vaginal Birth with a Perspective of Future Fertility:

A Discrete Choice Experiment

PONE-D-24-08282R1

Dear Dr. Crispin 

We’re pleased to inform you that your manuscript has been judged scientifically suitable for publication and will be formally accepted for publication once it meets all outstanding technical requirements.

Kind regards,

Natasha L Pritchard

Academic Editor

PLOS ONE

Additional Editor Comments (optional):

Dear authors

Thank you for addressing the revision points raised. They have been adequately addressed, and I thank you for your novel contribution to the literature on this important topic.
---

## [Editor Report · Acceptance letter]

5 Sep 2024

PONE-D-24-08282R1 

PLOS ONE

Dear Dr. Crispin, 

I'm pleased to inform you that your manuscript has been deemed suitable for publication in PLOS ONE. Congratulations! Your manuscript is now being handed over to our production team.

Kind regards, 

on behalf of

Dr. Natasha L Pritchard 

Academic Editor

PLOS ONE